# Biodiversity-Friendly Management in Olive Groves Supports Pollinator Conservation in a Mediterranean Terraced Landscape

**DOI:** 10.3390/insects16020198

**Published:** 2025-02-12

**Authors:** Matteo Dellapiana, Virginia Bagnoni, Laura Buonafede, Alice Caselli, Simone Marini, Malayka Samantha Picchi, Tiziana Sabbatini, Anna-Camilla Moonen

**Affiliations:** 1Institute of Plant Sciences, Sant’Anna School of Advanced Studies, Piazza Martiri della Libertà 33, 56127 Pisa, Italy; virginia.bagnoni@santannapisa.it (V.B.); alice.caselli18@gmail.com (A.C.); simone.marini.it@gmail.com (S.M.); maepik@gmail.com (M.S.P.); tiziana.sabbatini@santannapisa.it (T.S.); camilla.moonen@santannapisa.it (A.-C.M.); 2Department of Earth and Marine Sciences, University of Palermo, Via Archirafi, 22, 90123 Palermo, Italy; laura.buonafede@unipa.it; 3Department of Biotechnology and Bioscience, University of Milano-Bicocca, Piazza della Scienza, 2, 20126 Milano, Italy

**Keywords:** wild bees, butterflies, pan traps, habitat management, Mediterranean landscape, olive growing

## Abstract

This study explored the relationships between pollinator communities and habitats in traditional terraced olive groves in central Italy. The research assessed wild bee abundances, flowers, and butterfly communities across three habitat types: olive groves, herbaceous linear elements, and woody areas, using pan-traps and observation walks over two years. Results revealed the importance of habitat type over floral resources alone. Herbaceous linear elements and olive groves emerged as key contributors to pollinator diversity. These findings underscore the critical role that in-field semi-natural habitats can play in promoting pollinator-friendly agroecosystems and thus in conserving biodiversity and ecosystem services.

## 1. Introduction

Pollinators are essential contributors to agricultural ecosystems, enhancing the yield and quality of a wide range of crops, including those that are not directly dependent on animal pollination [1]. While crops like olives (*Olea europaea* L.) are primarily wind-pollinated [2], the presence of pollinators such as bees (Hymenoptera: Apoidea), butterflies (Lepidoptera: Rhopalocera), and other insects can still play a crucial role in maintaining the biodiversity of the surrounding ecosystems [3]. These pollinators support a variety of other plant species that coexist within agricultural landscapes, thereby ensuring the stability and resilience of these agroecosystems. In the Mediterranean region, where traditional agricultural practices have shaped the landscape for centuries [4], pollinators are recognized as a key element of the agroecosystem [5,6,7]. The Monte Pisano area in Tuscany, Italy, exemplifies this dynamic. Here, olive groves dominate the agricultural landscape [8], interspersed with herbaceous linear elements and woodland patches that create a mosaic of habitats. These diverse habitats, while not directly linked to olive production, are critical for supporting pollinator communities by providing nesting sites and diverse floral resources [9,10]. The olive-growing practices in Monte Pisano are typically non-intensive and occur on marginal lands characterized by steep terraced slopes and rocky soils. These small-scale, traditional farming practices have historically supported rich biodiversity, including a wide variety of pollinators [11,12]. However, the abandonment of these practices, coupled with the potential shift toward more intensive agricultural methods, poses a significant threat to the region’s biodiversity. Intensive agricultural practices have been recently shown to reduce bees’ α-diversity (i.e., local diversity within a site) consistently across taxa, while their effects on β-diversity (i.e., compositional differences between sites) and γ-diversity (i.e., overall diversity across the landscape) remain less clearly understood [13]. Investigating the specific interactions between pollinators and the various habitats within the olive-growing landscape of Mediterranean marginal land is important for developing policies that can inform conservation strategies and maintain the ecological integrity of this unique agricultural landscape. Over the years, European initiatives such as the Council Directive 92/43/EEC (21 May 1992) have recognized the critical role habitats play in maintaining biodiversity and ecosystem services within agricultural landscapes. One of the cornerstone frameworks in this effort is the High Nature Value farming (HNVf) concept, which focuses on preserving low-intensity farming systems that promote biodiversity through traditional, sustainable agricultural practices [14]. The latest iteration of the Common Agricultural Policy (2023–2027) introduced eco-schemes (ESc), a new mechanism designed to further integrate environmental sustainability into farming practices. Eco-schemes encourage farmers to adopt agricultural methods that enhance biodiversity, reduce chemical inputs, and promote sustainable land management [15]. These schemes offer financial incentives for practices such as agroecology, organic farming, and the maintenance of landscape features that support pollinators and other wildlife. Among these schemes, ESc 2 promotes the maintenance of ground herb cover in olive groves, a practice that has been proven to enhance pollinators [16], while ESc 5, which targets pollinators specifically, incentivizes practices that limit pesticide use, aligning olive-growing with the needs of pollinator conservation [17] and promotes the maintenance of pollinators-suitable habitats in the agroecosystem, up to 10% of the total landscape.

Research on habitats and pollinators in perennial crops highlights the critical role that diverse habitats within agricultural landscapes play in sustaining pollinator communities. In perennial systems such as olive groves, the availability of semi-natural habitats (SNHs) like woody patches, field margins, and inter-row cover crops positively influences pollinator diversity and abundance [18,19]. Maintaining permanent vegetation and minimizing soil disturbance [20] also benefits pollinators. The continuity of resources in these systems, compared to annual cropping systems, offers more stable foraging environments [21]. Maintaining herbaceous ground cover in olive groves has been linked to increased pollinator diversity [22]. Enhancing landscape complexity has shown positive effects on pollinator abundance [23]. Moreover, diversified farming systems enhance resilience against climate variability and promote ecological stability, benefiting pollinators in the long term [24,25]. In this study, we aimed to compare the contribution of olive groves that maintain a permanent spontaneous ground cover (OL) to the two key semi-natural habitats, herbaceous linear elements (HL) and woody area elements (WA), in their capacity to support two different groups of important pollinating insect: wild bees (Hymenoptera: Apoidea, excluding *Apis mellifera*, which was not considered due to the frequent movement of managed hives by beekeepers within the landscape) and butterflies (Lepidoptera: Rhopalocera). These two groups show different foraging behavior and have different life cycles and resource needs. It is therefore important to study the dependency of both groups on the three main SNHs and identify potential trade-offs to optimize habitat management at the landscape scale.

Previous studies have shown that flower availability and diversity, particularly in linear habitats, can influence pollinator abundance and diversity [26,27,28,29,30]. However, such interactions are less studied in Mediterranean agroecosystems, particularly in olive groves with permanent spontaneous vegetation. Given the importance of habitat diversity and floral resources for pollinators, we hypothesized that (i) hymenopteran pollinator abundance would be higher in habitats with greater floral abundance, particularly in HL elements, and that (ii) the abundance and composition of butterfly communities would vary significantly across habitats and sampling times, reflecting their specific phenological changes, and the shift in floral resources throughout the season, as it is suggested by previous research [31,32,33].

## 2. Materials and Methods

### 2.1. Study Area

The research was conducted in the traditional olive groves of the Monte Pisano region, located between the Arno and Serchio rivers in Tuscany’s Pisa province, central Italy (43.7217° N, 10.4942° E, WGS 84). The olive groves included in this study were distributed across two villages located 7 km apart: Calci and Vicopisano. Olive cultivation has significantly shaped the region since the 17th century [8], with most groves in the Monte Pisano area featuring ancient, terraced landscapes supported by dry stone walls. These terraces are essential for managing steep terrain, preventing soil erosion, and enhancing water retention [34]. The predominant form of agriculture in this area is smallholder, non-professional farming. The climate of the region is hot-summer Mediterranean, with an average annual rainfall of 1034.8 mm (SD = 252.7) between 2013 and 2022 (http://www.sir.toscana.it/, accessed on 20 August 2024, Uliveto Terme weather station), mostly in autumn and winter. In the area, an increase in extreme rainfall events has been reported [35]. In the two years of sampling, 2021 showed an average cumulated rainfall of 1042 mm, 0.03 SD above the mean, although January was markedly above the monthly average (287.8 mm, 1.39 SD above the mean for that month), while 2022 had a markedly low rainfall (594.2 mm, 1.74 SD below the mean) (Figure 1). Late spring and summer (May–September) both in 2021 and 2022 were characterized by lower precipitation compared to the decennial monthly average (2021: 149 mm, 1.17 SD below the mean; 2022: 97.8 mm, 1.65 SD below the mean).

We selected 12 squares, each measuring 1 km × 1 km, as sampling units for this study. These squares were defined using the Universal Transverse Mercator (UTM) system, a global coordinate system that divides the Earth into zones based on a transverse Mercator projection. Out of these 12 UTM squares, 6 were located in Calci and 6 in Vicopisano. Each UTM square contained at least one olive farm. All UTM squares were similar in their land-use classes and distribution. Within these squares, we identified and investigated three key habitat types: (i) olive groves (OL); (ii) herbaceous linear elements (HL), including farmland paths, dry-stone walls, or road verges; and (iii) woody areas elements (WA), which comprised woodlands with mixed woody vegetation (Figure 2). Pan traps were used to collect data on wild pollinators (Hymenoptera: Apoidea, excluding *Apis mellifera*) abundance. Transect walks were used to record butterfly community composition since their catch in pan traps was almost nil.

### 2.2. Pan Trap Sampling for Wild Bees and Flower Resource Monitoring

Pan traps were set according to Westphal et al. [36]: 432 plastic bowls (400 mL Pro-Pac, Vechta, Germany) were sprayed with UV-bright yellow, white, and blue paint (Sparvar Leuchtfarbe, Spray-Color GmbH, Merzenich, Germany), as these colors have been shown to attract a wide range of pollinator species, including wild bees [37]. A triad of bowls of the three different colors was mounted on a single wooden stick, forming a cluster, hereafter named a triplet. In each focal habitat, four triplets were set along a transect, 25 m apart to reduce competition among triplets [38]. Six sampling rounds were performed: in 2022, T1 on May 16–17, T2 on June 15–16, T3 on July 13–14, and T4 on September 8–9; and in 2021, T1 on May 17–18 and T3 on July 14–15. Traps were placed at vegetation height, filled with about 350 mL of water and a drop of commercial liquid laundry detergent to reduce surface tension, and left active for 24 h per round. Within a 2 m radius of each triplet, all flowers were identified and counted, apart from those belonging to the Graminaceae family. Yellow flowered species belonging to the Asteraceae family, Cichorieae tribe, exhibiting similar floral characteristics, were grouped and recorded as a single entry. Upon trap removal, trapped insects were extracted with the aid of coffee filters, and the filters containing the specimens were stored at −20 °C until sorting. Wild bees and bumblebees (Hymenoptera: Apoidea, excluding *Apis mellifera*) were separated, washed, and identified to the order level and counted. Data were pooled per trap regardless of the bowl’s color and aggregated at the habitat level for both hymenopteran pollinators and flowers (i.e., the sum of the four traps in each habitat). For the indicator species analysis, bee abundance was classified into four classes: “Absent” (0 individuals), “Low” (1–3 individuals), “Medium” (4–7 individuals), and “High” (>7 individuals).

### 2.3. Transect Walks for Butterfly Sampling

In each habitat in each UTM square, we defined a sampling transect of 100 m in length, and we surveyed diurnal butterflies (Lepidoptera: Rhopalocera, families: Nymphalidae, Pieridae, Hesperiidae, and Lycaenidae). Pan traps were placed along this same transect. We walked once in each subsection four times in 2022 (T1: 19–20 May; T2: 21–22 June; T3: 18–19 July; T4: 12–13 September) and three times in 2021 (T2: 3–16 June; T3: 8–15 July; T4: 20–23 September), after dawn and before dusk (the earliest was 7:30, the latest was 18:30). Transects were walked during optimal weather for butterfly sampling [39]. During each walk, an operator walked at a steady and constant pace of 10 m/min counting all the individual butterflies in an area of 2.5 m on either side, 5 m ahead, and 5 m above the ground. All the individuals were assigned to a species or a morpho-group if their identification in the field was not possible. Morpho-group samples were identified in the laboratory according to morphological keys [40,41].

### 2.4. Data Analysis

The effects of habitat type, time of sampling, total abundance of flowers, and Shannon diversity index (H’) of flowers on the abundance of the sum of wild bees and bumblebees were analyzed using a Generalized Linear Mixed Model (GLMM) with a negative binomial distribution to account for overdispersion in count data. The full model incorporated fixed effects for every explanatory variable and all possible interactions, along with a random effect for the quadrant of sampling to account for spatial variation and a random effect for the year to account for inter-annual variability.

Another GLMM based on the negative binomial distribution focused on the abundance of flowers as a response variable. The full model incorporated fixed effects for habitat and time of sampling, with their interaction, along with a random effect for the quadrant of sampling to account for spatial variation and a random effect for the year to account for inter-annual variability.

A third GLMM based on the negative binomial distribution focused on butterfly abundance. The full model incorporated habitat, year, and time of sampling with their interactions as fixed effects and sampling quadrant as a random effect, as the inclusion of sampling year as a random effect led to singularity issues in the fit, probably due to the limited dimensions of the dataset.

For each model, the significance of predictors was assessed using Wald Type II Chi-squared tests, and model selection was guided by Akaike Information Criterion (AIC). Each non-significant variable or interaction was removed from the full model, up to the point at which only significant variables remained and AIC no longer decreased. The validity of model assumptions was confirmed through residual diagnostics. Estimated Marginal Means (EMMs) were computed for significant predictors, and pairwise comparisons were conducted using Hochberg-adjusted *p*-values.

Permutational multivariate analysis of variance (PERMANOVA) was employed to examine the effects of habitat and time of sampling on flower and butterfly community composition, using Bray–Curtis dissimilarity, 9999 permutations, and year of sampling as a stratum to restrict permutations and to account for inter-annual variability. Pairwise comparisons between groups were performed with Hochberg *p*-value adjustment. Non-metric multidimensional scaling (NMDS) was utilized to visualize patterns in community composition using Bray–Curtis dissimilarity with three dimensions (k = 3) and a maximum of 100 iterations.

Indicator species analysis (ISA) was conducted to identify plant species significantly associated with specific habitats or pollinator abundance classes, and butterfly species significantly associated with specific habitats or times of sampling. Significance was tested with 999 permutations.

All analyses were conducted in R (version 4.3.1) [42], utilizing the ‘glmmTMB’ (version 1.1.10) [43], ‘DHARMa’ (version 0.4.7) [44], ‘emmeans’ (version 1.10.16) [45], ‘indicspecies’ (version 1.7.15) [46], ‘multcomp’ (version 1.4-26) [47], ‘pairwiseAdonis’ (version 0.4.1) [48], ‘car’ (version 3.1-3) [49]. Plots were created with the ‘ggplot2’ (version 3.5.1) [50], ‘ggpubr’ (version 0.6.0) [51], and ‘ggforce’ (version 0.4.2) [52] packages. Figures in Appendix A were created with the ‘lattice’ (version 0.21-8) [53] package.

## 3. Results

### 3.1. Wild Bees

A total of 471 wild bees and bumblebees (Hymenoptera: Apoidea, excluding *Apis mellifera*) were collected with pan traps summing both years and all times of samplings (Appendix A), 128 in 2021 and 343 in 2022. The habitat in which most specimens were found was HL (mean considering all four traps and both years and all sampling times = 3.06, sd = 4.35) followed by OL (mean = 2.35, sd = 2.90) and WA (mean = 1.61, sd = 1.69). The highest average abundance per habitat was observed at T2 (mean considering all four traps = 4.24, sd = 5.72), followed by T3 (mean = 2.36, sd = 2.25) and T1 (mean = 2.22, sd = 2.65), with the lowest abundance recorded at T4 (mean = 0.73, sd = 0.84).

The analysis performed with GLMM (Table 1) aimed at investigating the role of habitat and flower availability and diversity on pollinator captures, which indicated several patterns in pollinators abundance: the most important variable in the model was time of sampling (Type II Wald test Chi-Squared = 31.95, *p* = 5.35 × 10^−7^), followed by habitat of sampling (Type II Wald test Chi-Squared = 7.68, *p* = 0.02). No significant effects were observed for flower abundance and flower Shannon diversity index.

Pairwise comparisons between times of sampling (Figure 3b) showed that T1 and T4 differed significantly (EMM ratio = 3.34, SE = 1.01, *p* < 3.00 × 10^−4^), as well as T2 and T4 (EMM ratio = 5.29, SE = 1.58, *p* < 1.00 × 10^−4^) and T3 and T4 (EMM ratio = 3.66, SE = 1.07, *p* = 1.00 × 10^−4^). The only significant pairwise comparison between habitats (Figure 3a) highlighted the difference between HL and WA, where HL had a higher count than WA (EMM ratio = 1.72, SE = 0.34, *p* = 0.02).

### 3.2. Flowers

A total of 34,978 individual flowers belonging to 109 species (Appendix A) were sampled across the two years and all times of sampling, 10,685 in 2021 and 24,293 in 2022. The highest flower abundance per habitat was recorded at T1 (mean considering both years = 490.66, sd = 426.06), followed by T2 (mean = 82.48, sd = 209.63), T3 (mean = 44.22, sd = 89.30), and T4 (mean = 22.64, sd = 39.13). The habitat in which most flowers were found was the olive grove (mean considering both years and all sampling times = 272.64, sd = 476.74), followed by HL (mean = 157.15, sd = 279.19) and WA (mean = 80.26, sd = 172.79).

The GLMM that was focused on the influence of habitat and time of sampling on total flower abundance highlighted a significant effect of both variables (Table 2). Time of sampling (Type II Wald test Chi-Squared = 96.62, *p* = 2.2 × 10^−16^) was the most important predictor. Pairwise comparisons (Figure 4b) showed significant differences between T1 and T2 (EMM ratio = 3.81, SE = 0.83, *p* < 1.00 × 10^−4^), T1 and T3 (EMM ratio = 5.32, SE = 0.99, *p* < 1.00 × 10^−4^), and T1 and T4 (EMM ratio = 6.8, SE = 1.69, *p* < 1.00 × 10^−4^). Habitat (Type II Wald test Chi-Squared = 38.29, *p* = 4.84 × 10^−9^) was also an important predictor of flower abundance. Pairwise comparisons highlighted significantly different abundances across all habitats (Figure 4a): OL and HL (EMM ratio = 0.73, SE = 0.12, *p* = 0.05), OL and WA (EMM ratio = 3.33, SE = 0.65, *p* < 1.00 × 10^−4^), and HL and WA (EMM ratio = 2.43, SE = 0.48, *p* < 1.00 × 10^−4^).

To assess the differences in flower community composition across habitats and times of sampling, we performed a PERMANOVA on the community Bray–Curtis distance matrix. The results (Table 3) revealed that both habitat (R^2^ = 0.02, F = 1.40, *p* = 0.04) and, most importantly, time of sampling (R^2^ = 0.59, F = 3.17, *p* < 1 × 10^−4^) significantly influenced the composition of flower communities. When comparing flower communities across different habitats, the PERMANOVA results indicated a significant difference between OL and WA habitats (R^2^ = 0.02, F = 1.85, *p* = 0.01). However, no significant differences were found between HL elements and OL (R^2^ = 0.01, F = 1.11, *p* = 0.34), nor between HL and WA (R^2^ = 0.01, F = 1.25, *p* = 0.17). The effect of time on flower community composition was robust, with significant differences observed between T1 and T3 (R^2^ = 0.04, F = 5.43, *p* = 1 × 10^−4^), T1 and T2 (R^2^ = 0.04, F = 3.40, *p* = 1 × 10^−3^), T1 and T4 (R^2^ = 0.05, F = 4.29, *p* = 1 × 10^−4^), and between T2 and T4 (R^2^ = 0.04, F = 1.60, *p* = 0.01). These findings are visually supported by the Non-Metric Multidimensional Scaling (NMDS) plots, which illustrate the clustering of flower communities according to the time of sampling (Figure 5a) and habitat (Figure 5b).

Time of sampling showed a significant alignment with the NMDS axes (R^2^ = 0.09, *p* = 0.001). Centroids for different sampling times indicated a clear temporal gradient in community composition, with distinct shifts between T1 (−0.2006, 0.0676) and T4 (0.2206, −0.1082). The stress value was 0.221.

The Indicator Species Analysis showed that *Passiflora caerulea* and *Scabiosa triandra* are significantly correlated with high pollinator abundance. This analysis also identified specific flower species that are strongly associated with habitats (Table 4). *Anthemis arvensis* and *Potentilla reptans* were strongly associated with HL habitats. In contrast, *Hypericum perforatum*, *Solanum nigrum*, and *Centaurium erythraea* were significant indicators of OL habitat, while *Dorycnium hirsutum* was associated with WA habitats.

### 3.3. Butterflies

A total of 1023 butterfly specimens belonging to 49 species were observed (Appendix A), 505 in 2021 and 518 in 2022. The most butterfly-rich habitat was OL (mean considering both years and all sampling times = 4.80, SD = 2.25), followed by HL (mean = 4.31, SD = 2.04) and WA (mean = 4.11, SD = 2.55). The average butterfly abundance was the highest in the second sampling time (mean considering all habitats and both years = 8.18, sd = 7.00), with the lowest abundance recorded in the third sampling time (mean = 2.23, sd = 2.86). The only significant predictor was sampling time (Type II Wald Chi-Square = 59.532, *p* = 7.40 × 10^−13^) in the GLMM (Table 5). Pairwise comparisons between the sampling times (Figure 6b) highlighted significant differences between T1 and T2 (EMM ratio = 0.50, SE = 0.10, *p* = 8 × 10^−3^), T2 and T3 (EMM ratio = 2.91, SE = 0.49, *p* < 1 × 10^−4^), T2 and T4 (EMM ratio = 2.83, SE = 0.48, *p* < 1 × 10^−4^). Habitat type had no significant effect on butterfly abundance (Figure 6a).

The PERMANOVA analysis (Table 6) revealed significant differences in butterfly community composition across both habitats and sampling times (Habitat: F = 1.58, *p* = 1.90 × 10^−2^; Time: F = 5.51, *p* = 1 × 10^−4^). Pairwise comparisons showed that the most substantial differences were observed between the OL and WA habitats (F = 2.32, *p* = 3 × 10^−3^) and between HL and WA (F = 1.82, *p* = 1.50 × 10^−2^), whereas the differences between HL and OL were not significant. Similarly, significant differences in community composition were found between all sampling times, further underscoring the influence of temporal factors on butterfly community dynamics. These findings are visually supported by the NMDS plots (Figure 7) which illustrate the clustering of butterflies’ communities according to habitat and time.

Habitat and time of sampling both showed significant alignments with the NMDS axes. Habitat explained 3.7% of the variation in community composition (R^2^ = 0.037, *p* = 0.01). Centroids indicated subtle differentiation among habitats, with HL (−0.028, 0.015), OL (0.0281, 0.083), and WA (−0.0008, −0.1241). Time of sampling explained a more substantial portion of the variation, accounting for 14.5% of the community composition (R^2^ = 0.1447, *p* = 0.001). Centroids for different sampling times revealed a temporal gradient, with notable shifts from T1 (−0.0139, 0.2717) to T4 (0.1173, −0.0245). The stress value was 0.245.

Additionally, the Indicator Species Analysis revealed that *Limenitis reducta* and *Pararge aegeria* were significantly associated with the WA habitat (*p* = 0.003 and *p* = 0.013, respectively). Several significant associations between butterfly species and times of sampling were observed as well (Table 7).

## 4. Discussion

In this study, we highlight the contribution of olive groves with a permanent ground cover of spontaneous vegetation in supporting the hymenopteran and lepidopteran pollinator communities. Abundances of both groups in the low-input olive groves were similar to the ones registered in the semi-natural habitats in the landscape.

The abundance of wild bees and bumblebees was significantly higher in the herbaceous linear (HL) habitat compared to the woody area (WA). The intermediate abundance observed in the olive grove (OL) habitat, despite having the highest flower abundance, suggests that factors beyond floral resources, such as nesting opportunities or microclimate, may influence pollinator abundance. Alternatively, this pattern may be explained by the fact the olive groves cover a much higher surface than the linear habitats, and pollinator abundances are therefore more diluted. These findings are consistent with previous studies that emphasize the complexity of habitat preferences for wild bees, where the availability of nesting sites, microclimatic conditions, and habitat connectivity can play crucial roles alongside floral resources [11,54,55,56,57,58]. Previous research has also highlighted that wild bee abundance is directly proportional to the diversity of flower resources [26,28,59,60], while in this study, no direct effect of the flower Shannon diversity index was observed. We observed that *Passiflora caerulea* and *Scabiosa triandra* are significantly associated with a high abundance of hymenopteran pollinators. Both species have been recognized as entomophilous [61,62], and they both present purple-pink flowers that are thought to be attractive to bees [63].

As the interaction between habitat and time of sampling was non-significant in the model, the trend in wild bee and bumblebee abundances in different habitats was consistent across times. In the first year, the sampling design included only two sampling times (T1: May and T3: July) for the pan traps, which may have reduced the temporal resolution of data on seasonal dynamics. However, this was mitigated in the second year by increasing sampling events, providing a more complete dataset. The temporal mismatch observed in 2022 between peak flower abundance (T1: May) and peak pollinator abundance (T2: June) suggests a potential phenological mismatch [64] irrespective of the lower precipitations observed in June compared to May in both years, as higher precipitations have been shown to correlate with higher pollinator abundance and activity [65,66]. The influence of management practices in the OL habitat, such as mowing, could also contribute to this mismatch. This practice may disrupt the natural flowering cycles and reduce habitat quality, particularly during critical periods when pollinators are most active. Interestingly, previous works highlighted both a lower abundance of wild bees in managed orchards compared to any other habitat in the landscape [21,67] and a higher abundance in managed olive groves with respect to woody areas [5]. These contradictory results point to a context-specific pattern in population dynamics. The HL habitat, characterized by its connectivity, appears to play a key role in supporting the pollinator community. In contrast, the more homogeneous and shadier WA habitat, with its limited floral diversity and abundance, may provide fewer resources for pollinators, leading to lower abundance. It is important to note however that several studies have underlined the importance of small forest patches within agricultural landscapes for the pollinator community [68,69,70]. These results have important implications for the management of traditional olive groves in the Mediterranean region. Enhancing habitat diversity through the preservation of herbaceous linear elements and careful management of olive groves could be key strategies for promoting pollinator abundance and diversity, as pointed out by earlier research [19,22].

Finally, the choice of specific pan trap colors and observation protocols was aligned with established practices [36]. However, some concerns have been raised about the efficacy of pan traps when evaluating habitats with large flower density gradients [71]. These methodological nuances underline the need for cautious interpretation of abundance data, emphasizing relative rather than absolute trends.

The results from the GLMM indicate a clear temporal pattern in butterfly abundance, with T2 showing a higher abundance than any other time of sampling, as was the case for wild bees and bumblebees, potentially due to optimal weather, food availability, or other ecological factors that peak during this period. This pattern aligns with the general understanding of butterfly ecology, where abundance peaks often occur in response to specific conditions, both resource (food, nesting) and biotope-related [72,73]. However, while total butterfly abundance varied by time, the PERMANOVA analysis revealed significant differences in butterfly community composition across all habitats and sampling times. The most substantial habitat-related differences were observed between OL and WA, and between HL and WA, suggesting that while the overall number of butterflies may not differ drastically between habitats, the species composition does. This indicates that different habitats support different butterfly communities [74], likely due to variations in floral resources, microclimate, and other ecological factors that influence habitat suitability for different butterfly species [75]. The ISA further highlighted the relationship between specific butterfly species and their preferred habitats and times of activity. Notably, *Limenitis reducta* and *Pararge aegeria* were significantly associated with the WA habitat. This association underscores the importance of WA habitat in supporting these butterfly species, possibly due to factors like shade, moisture, or specific host plants available in these areas [76]. These findings suggest that while butterflies in the study area may be somewhat generalist in terms of habitat use, the specific community composition is still strongly influenced by the habitat type.

These results highlight that conservation strategies should focus on maintaining and enhancing habitats, not only targeting specific insect or plant species. Garibaldi et al. [77] suggested that at least 20% of the agricultural landscape should consist of SNHs to support ecosystem services, particularly regulating services such as pollination and pest control [78]. Enhancing botanical diversity within agricultural systems is also crucial. Wood et al. [54] highlighted the importance of including a variety of plant species in agri-environmental schemes, to foster pollinator diversity and abundance. Other studies show that semi-natural grasslands and herbaceous elements are critical for maintaining pollinator species richness and abundance due to their high floral availability, microhabitat diversity, and the ecological stability they provide [79,80]. Furthermore, these herbaceous elements often function as transitional habitats at orchard edges or roadside borders, contributing to landscape connectivity [81]. Our study enforces the suggestion that biodiversity-friendly management in perennial woody crops can provide a crucial contribution to biodiversity conservation [82,83]. In this case, the maintenance of a permanent vegetation cover in olive groves proves to be a crucial element in terraced landscapes. Although the study area contains a high amount of woodland, this habitat alone is not able to support diverse butterfly communities and abundant pollinator communities. The olive groves instead provide a huge surface able to sustain unique butterfly communities and abundant wild bee communities. They are similar to herbaceous linear elements, but it should be noted that the total surface of olive groves outweighs the surface of the linear elements and therefore at a landscape scale, this in-field semi-natural habitat provides a fundamental contribution to butterfly and wild bee community conservation.

The study was conducted in two villages, Calci and Vicopisano, located 7 km apart from one another. Although location effects were implicitly accounted for in the model through random effects, the species composition of flowers, bees, and butterflies between these villages was not explicitly analyzed. Future work could benefit from a detailed assessment of these spatial differences, as their inclusion could strengthen the ecological validity of the data and highlight site-specific conservation needs. Furthermore, the exposition of the two areas is slightly different, and this causes differences in micro-climatic conditions. Despite these differences, the two areas show a coherent response of butterfly and wild bee communities to semi-natural habitats.

## 5. Conclusions

This study highlights the complex relationships between habitat structure, floral abundance, and pollinator dynamics in Mediterranean olive-growing landscapes. Despite olive groves exhibiting the highest flower abundance, it was the herbaceous linear elements that supported the greatest abundance of bee-like pollinators. This suggests that factors beyond floral availability, such as habitat structure and connectivity, play a crucial role in supporting the pollinator community. The linear arrangement and edge characteristics of these elements likely create diverse microhabitats and improve habitat connectivity, making them more attractive to pollinators. The temporal mismatch observed between peak floral abundance and peak pollinator activity further emphasizes the need to account for phenological cycles and the potential impact of land management practices like mowing on pollinator resources. The study also demonstrated that different habitats support distinct flower and butterfly communities, with the most marked differences observed between olive groves and woody areas. Specific butterfly species were found to rely on different habitats, illustrating the importance of maintaining habitat diversity to support a wide range of species. Overall, this study reinforces the idea that conservation strategies should focus on preserving and enhancing habitat structures within agricultural landscapes. Instead of only targeting individual species, efforts should prioritize maintaining the integrity and connectivity of habitats, ensuring that they provide the necessary resources for pollinators throughout their life cycles. This habitat-based approach is key to promoting long-term biodiversity and ecosystem resilience in Mediterranean agricultural systems.

## Figures and Tables

**Figure 1 insects-16-00198-f001:**
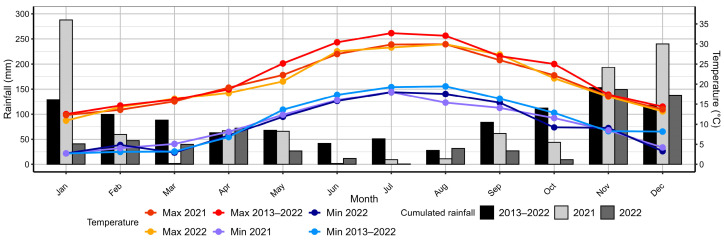
Monthly cumulated rainfall and average minimum and maximum monthly temperature (maximum and minimum) Uliveto Terme weather station, from January 2013 to December 2022.

**Figure 2 insects-16-00198-f002:**
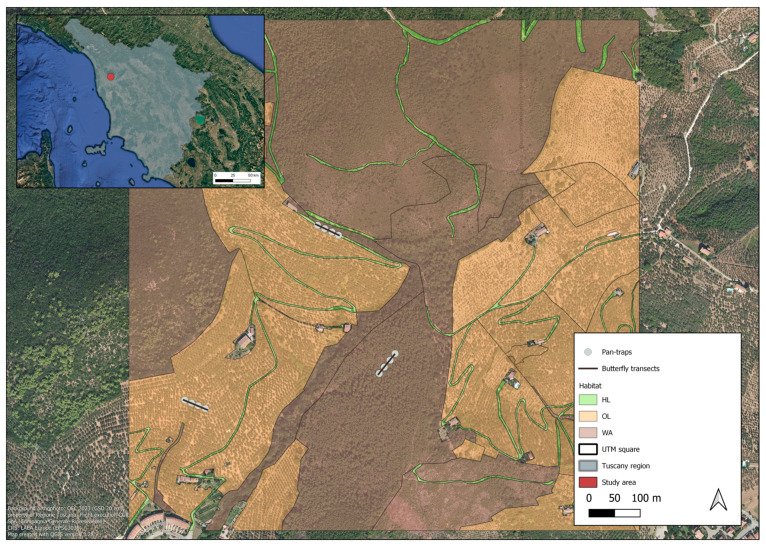
Inset map: location of the study area in Central-Western Italy. Main canvas: example of a UTM 1 km^2^ study unit, showing the sampling setup in the three different habitats.

**Figure 3 insects-16-00198-f003:**
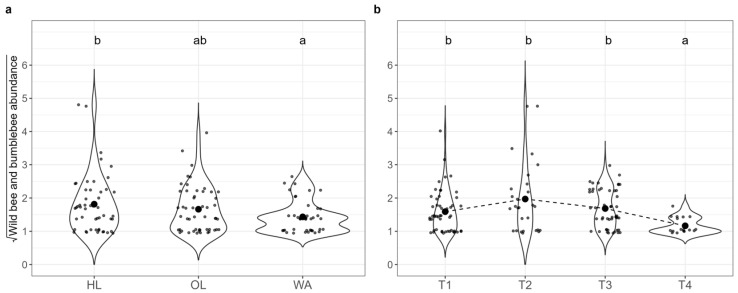
Abundance of wild bees and bumblebees (square root transformed) in both years (2021 and 2022 aggregated) across the three habitats: (**a**) (HL: herbaceous linear, OL: olive grove, WA: woody area); and four sampling times: (**b**) (T1: May, T2: June, T3: July, T4: September). Significant differences between habitats and sampling times are denoted by different letters (*p* < 0.05, with Hochberg’s adjustment). Large black dots in panels (**a**,**b**) represent the means for each group.

**Figure 4 insects-16-00198-f004:**
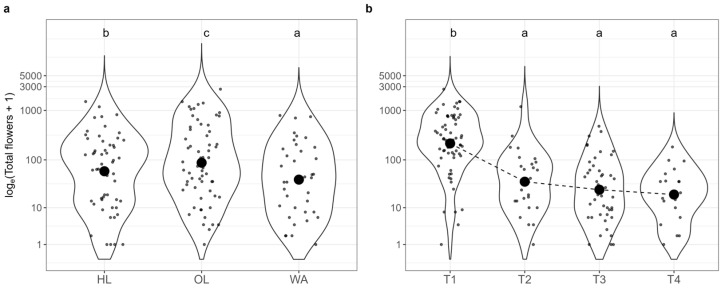
Abundance of flowers (log-transformed) in both years (2021 and 2022 aggregated) across the three habitats: (**a**) (HL: herbaceous linear, OL: olive grove, WA: woody area); and four sampling times: (**b**) (T1: May, T2: June, T3: July, T4: September). Significant differences between habitats and sampling times are denoted by different letters (*p* < 0.05, with Hochberg’s adjustment). Large black dots in panels (**a**,**b**) represent the means for each group.

**Figure 5 insects-16-00198-f005:**
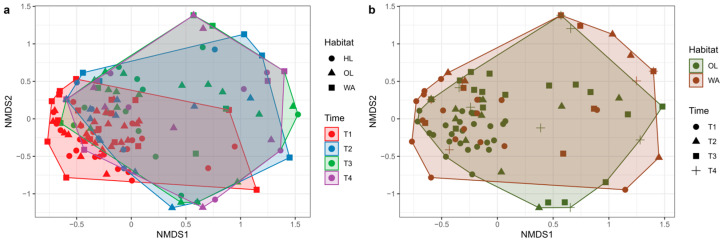
NMDS plots showing the differences in flower community composition across sampling times (**a**) and habitats (**b**). The analysis was based on the Bray–Curtis dissimilarity matrix, calculated using the square-root transformed species abundance data. NMDS stress value = 0.221. Panel (**a**): shapes represent the different habitats: herbaceous linear (HL, circles), olive grove (OL, triangles), and woody area (WA, squares). Colors represent the four sampling times: May (T1: red), June (T2: blue), July (T3: green), and September (T4: purple). Convex hulls illustrate the grouping of communities within each time. Panel (**b**): shapes represent the different sampling times: T1 (circles), T2 (triangles), T3 (squares), and T4 (crosses). Colors indicate the two habitats: olive grove (OL, green) and woody area (WA, brown). Herbaceous linear (HL) is not shown to ease interpretation. Convex hulls illustrate the separation of communities within each habitat.

**Figure 6 insects-16-00198-f006:**
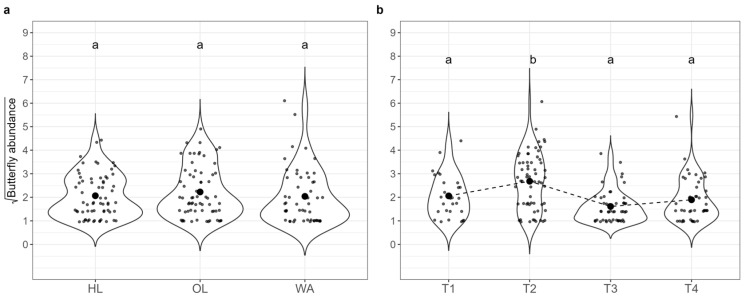
Butterfly abundance (square root transformed) in both years (2021 and 2022 aggregated) across the three habitats: (**a**) (HL: herbaceous linear, OL: olive grove, WA: woody area); and four sampling times: (**b**) (T1: May, T2: June, T3: July, T4: September). Significant differences between sampling times are indicated by different letters (*p* < 0.05 with Hochberg’s adjustment). The black circles represent the mean butterfly abundance.

**Figure 7 insects-16-00198-f007:**
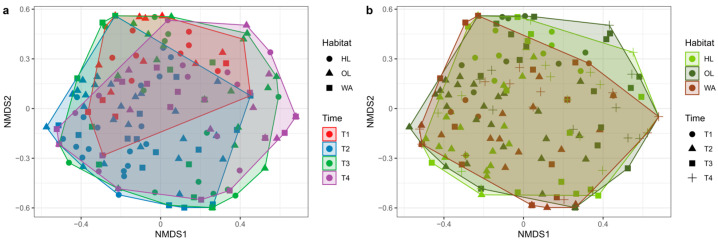
NMDS plots showing the differences in butterfly community composition across sampling times (**a**) and habitats (**b**). The analysis was based on the Bray–Curtis dissimilarity matrix, calculated using the square-root transformed species abundance data. NMDS stress value = 0.245. A: shapes represent the different habitats: herbaceous linear (HL, circles), olive grove (OL, triangles), and woody area (WA, squares). Colors represent the four sampling times: May (T1: red), June (T2: blue), July (T3: green), and September (T4: purple). Convex hulls illustrate the grouping of communities within each time. B: shapes represent the different sampling times: T1 (circles), T2 (triangles), T3 (squares), and T4 (crosses). Colors indicate the three habitats: HL (light green), OL (dark green), and WA (brown). Convex hulls illustrate the separation of communities within each habitat.

**Table 1 insects-16-00198-t001:** Model parameters for wild bee and bumblebee abundances. Fixed effects include the intercept, habitats (HL: herbaceous linear as reference, OL: olive grove, and WA: woody area), and sampling times (T1: May as reference). Dispersion and random effects are also reported. Estimates are presented with 95% confidence intervals (CI) and *p*-values.

Parameter	Estimate (CI)	*p*
Fixed Effects		
(Intercept)	2.67 (1.78, 3.98)	<0.001
Habitat (OL)	0.82 (0.58, 1.16)	0.264
Habitat (WA)	0.58 (0.40, 0.85)	0.006
Time (T2)	1.59 (0.99, 2.54)	0.055
Time (T3)	1.09 (0.76, 1.57)	0.623
Time (T4)	0.30 (0.17, 0.54)	<0.001
Dispersion		
(Intercept)	1.68 (1.13, 2.49)	
Random Effects		
SD (Intercept: Landscape)	0.28 (0.13, 0.60)	
SD (Intercept: Year)	0.13 (0.02, 0.96)	

**Table 2 insects-16-00198-t002:** Model parameters for flower abundance. Fixed effects include the intercept, habitats (HL: herbaceous linear as reference, OL: olive grove, and WA: woody area), and sampling times (T1: May as reference). Dispersion and random effects are also reported. Estimates are presented with 95% confidence intervals (CI) and *p*-values.

Parameter	Estimate (CI)	*p*
Fixed Effects		
(Intercept)	385.16 (277.58, 534.43)	<0.001
Habitat (OL)	1.37 (1.00, 1.87)	0.051
Habitat (WA)	0.41 (0.28, 0.61)	<0.001
Time (T2)	0.26 (0.16, 0.43)	<0.001
Time (T3)	0.19 (0.13, 0.27)	<0.001
Time (T4)	0.14 (0.08, 0.26)	<0.001
Dispersion		
(Intercept)	329.03 (242.50, 446.43)	
Random Effects		
SD (Intercept: Landscape)	0.24 (0.10, 0.54)	
SD (Intercept: Year)	0.04 (3.85 × 10^−7^, 3380.13)	

**Table 3 insects-16-00198-t003:** PERMANOVA results and significant pairwise comparisons for flower community composition across different habitats (herbaceous linear: HL, olive grove: OL, woody area: WA) and times of sampling (T1: May, T2: June, T3: July, T4: September). The analysis was based on a Bray–Curtis dissimilarity matrix. Year of sampling was used as a stratum for the permutations (n = 9999). Estimates include degrees of freedom (df), sum of squares, R^2^, F-statistic, and *p*-values (with Hochberg’s adjustment for pairwise comparisons).

Response Variable	Explanatory Variables	df	Sum of Squares	R^2^	F	*p*
Flowers community abundances, Bray–Curtis distance matrix	Habitat	2	1.154	0.017	1.395	0.042
	Time	3	3.931	0.059	3.173	0.0001
	Residual	149	61.618	0.924		
	Total	154	66.702	1		
Pairwise comparisons						
Habitat: HL vs. OL	Habitat	1	0.441	0.009	1.107	0.337
	Time	3	4.29	0.087	3.586	0.001
	Residual	112	44.66	0.904		
	Total	116	49.39	1		
Habitat: HL vs. WA	Habitat	1	0.532	0.013	1.25	0.168
	Time	3	2.392	0.059	1.875	0.002
	Residual	88	37.425	0.928		
	Total	92	40.35	1		
Habitat: OL vs. WA	Habitat	1	0.777	0.018	1.854	0.015
	Time	3	2.516	0.058	2.001	0.0001
	Residual	95	39.82	0.924		
	Total	99	43.11	1		
Time: T1 vs. T2	Time	1	1.348	0.037	3.396	0.001
	Habitat	2	1.212	0.033	1.526	0.041
	Residual	86	34.142	0.9303		
	Total	89	36.701	1		
Time: T1 vs. T3	Time	1	2.186	0.048	5.428	0.001
	Habitat	2	1.249	0.0273	1.551	0.015
	Residual	105	42.287	0.925		
	Total	108	45.722	1		
Time: T1 vs. T4	Time	1	1.698	0.049	4.29	0.0001
	Habitat	2	1.616	0.046	2.042	0.0007
	Residual	80	31.66	0.905		
	Total	83	34.974	1		
Time: T2 vs. T4	Time	1	0.705	0.035	1.599	0.011
	Habitat	2	0.844	0.042	0.957	0.493
	Residual	42	18.508	0.923		
	Total	45	20.057	1		

**Table 4 insects-16-00198-t004:** Results from the ISA showing species significantly associated with specific habitats and pollinator abundance classes. Species are listed for herbaceous linear (HL), olive grove (OL), woody area (WA), and transects with high pollinator abundance.

Group	Species (Family)	Stat	*p*
Habitat: HL	*Anthemis arvensis* (Asteraceae)	0.357	0.002
	*Potentilla reptans* (Rosaceae)	0.349	0.026
Habitat: OL	*Hypericum perforatum* (Hypericaceae)	0.378	0.04
	*Solanum nigrum* (Solanaceae)	0.366	0.01
	*Centaurium erythraea* (Gentianaceae)	0.306	0.035
Habitat: WA	*Dorycnium hirsutum* (Fabaceae)	0.229	0.047
Pollinator abundance: high	*Passiflora caerulea* (Passifloraceae)	0.311	0.043
	*Scabiosa triandra* (Caprifoliaceae)	0.284	0.049

**Table 5 insects-16-00198-t005:** Model parameters for butterfly abundances. Fixed effects include the intercept and sampling times (T1 as reference). Dispersion and random effects are also reported. Estimates are presented with 95% confidence intervals (CI) and *p*-values.

Parameter	Estimate (CI)	*p*
Fixed Effects		
(Intercept)	4.38 (3.13, 6.13)	<0.001
Time (T2)	1.77 (1.23, 2.55)	0.002
Time (T3)	0.61 (0.40, 0.92)	0.018
Time (T4)	0.63 (0.41, 0.95)	0.028
Dispersion		
(Intercept)	3.99 (2.96, 5.38)	
Random Effects		
SD (Intercept: Landscape)	0.16 (0.05, 0.048)	

**Table 6 insects-16-00198-t006:** PERMANOVA results and pairwise comparisons for butterfly community composition across different habitats (Herbaceous Linear (HL), Olive Grove (OL), Woody Area (WA), and times of sampling (T1: May, T2: June, T3: July, T4: September). The analysis was based on a Bray–Curtis dissimilarity matrix. Year of sampling was used as a stratum for the permutations (n = 9999). Estimates include degrees of freedom (df), sum of squares, R^2^, F-statistic, and *p*-values with Hochberg’s adjustment.

	Explanatory Variables	df	Sum of Squares	R^2^	F	*p*
Butterfly abundances, Bray–Curtis distance matrix	Habitat	2	1.245	0.016	1.579	0.019
	Time	3	6.514	0.084	5.509	0.0001
	Residual	177	69.763	0.900		
	Total	182	77.522	1.000		
Pairwise comparisons						
Habitat: HL vs. OL	Habitat	1	0.248	0.005	0.650	0.804
	Time	3	5.940	0.109	5.178	0.0001
	Residual	126	48.180	0.886		
	Total	130	54.369	1.000		
Habitat: HL vs. WA	Habitat	1	0.733	0.015	1.821	0.015
	Time	3	4.109	0.082	3.403	0.0001
	Residual	112	45.087	0.903		
	Total	116	49.929	1.000		
Habitat: OL vs. WA	Habitat	1	0.928	0.019	2.321	0.003
	Time	3	4.061	0.081	3.386	0.001
	Residual	113	45.177	0.901		
	Total	117	50.166	1.000		
Time: T2 vs. T1	Time	1	2.515	0.070	6.837	0.0001
	Habitat	2	1.491	0.041	2.027	0.002
	Residual	87	31.995	0.889		
	Total	90	36.001	1.000		
Time: T2 vs. T3	Time	1	2.196	0.048	5.528	0.0001
	Habitat	2	0.996	0.022	1.253	0.118
	Residual	107	42.515	0.930		
	Total	110	45.707	1.000		
Time: T2 vs. T4	Time	1	2.706	0.063	7.075	0.0001
	Habitat	2	0.987	0.023	1.29	0.093
	Residual	103	39.386	0.914		
	Total	106	43.078	1.000		
Time: T1 vs. T3	Time	1	1.249	0.039	3.027	0.0002
	Habitat	2	0.902	0.028	1.093	0.291
	Residual	72	29.708	0.932		
	Total	75	31.860	1.000		
Time: T1 vs. T4	Time	1	2.632	0.0874	6.724	0.0001
	Habitat	2	0.858	0.029	1.097	0.327
	Residual	68	26.614	0.884		
	Total	71	30.104	1.000		
Time: T3 vs. T4	Time	1	1.631	0.042	3.894	0.0001
	Habitat	2	0.630	0.016	0.752	0.782
	Residual	88	36.867	0.942		
	Total	91	39.128	1.000		

**Table 7 insects-16-00198-t007:** Results from the ISA showing species significantly associated with specific habitats and times of sampling. Stat represents the strength of association, and *p*-values indicate the level of significance.

Group	Species (Family)	Stat	*p*
Habitat: WA	*Limenitis reducta* (Nymphalidae)	0.326	0.006
	*Pararge aegeria* (Nymphalidae)	0.324	0.015
Time: T1	*Vanessa cardui* (Nymphalidae)	0.532	0.001
	*Coenonympha pamphilus* (Nymphalidae)	0.372	0.02
Time: T2	*Melanargia galathea* (Nymphalidae)	0.577	0.001
	*Colias crocea* (Pieridae)	0.555	0.001
	*Melitaea didyma* (Nymphalidae)	0.346	0.023
	*Pieris mannii* (Pieridae)	0.283	0.029
Time: T3	*Pontia edua* (Pieridae)	0.325	0.01
	*Pyronia cecilia* (Nymphalidae)	0.323	0.009
Time: T4	*Hipparchia statilinus* (Nymphalidae)	0.581	0.001
	*Limenitis reducta* (Nymphalidae)	0.312	0.012
	*Ochlodes sylvanus* (Hesperiidae)	0.302	0.006
	*Leptotes pirithous* (Lycaenidae)	0.302	0.01
	*Hipparchia fagi* (Nymphalidae)	0.278	0.023

## Data Availability

The data presented in this study are available as Appendix A.

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
