# Peer review of "Biodiversity-Friendly Management in Olive Groves Supports Pollinator Conservation in a Mediterranean Terraced Landscape"

_insects, 2025, doi:10.3390/insects16020198_

Round 1
Reviewer 1 Report
Comments and Suggestions for Authors
This well-planned study looked at how pollinator communities interact with habitats in terraced olive groves in central Italy. The two-year study found that habitat structure matters more than just floral abundance, with herbaceous elements playing a key role in improving pollinator diversity. The findings highlight the need for conservation strategies which improve habitat diversity and connectivity. While I appreciate the effort of the authors in addressing an important ecological issue, I believe there’s a lot of room to refine this work to make it feel more like a research paper rather than a thesis chapter. My comments are:
L1-4: Consider revising the title to make it clear and concise.
L16-26: It is recommended to add pan traps and observation walks as methods to improve the summary.
L38-40: Do you want to say analysis using GLMMs?
L27-48: Check the maximum number of words allowed as per the MDPI format and make it according to the guidelines.
L49: Expand the keywords and consider including the region and farming system.
L54: Add family.
L58: Add some recent papers here. For example:
1. Straub, F., Birkenbach, M., Boesing, A. L., Manning, P., Olsson, O., Kuppler, J., ... & Ayasse, M. (2025). Local and landscape factors differently influence health and pollination services in two important pollinator groups. Science of The Total Environment, 959, 178330.
2. Ali, Q., Ali, M., Khan, F. Z. A., Noureldeen, A., Alghamdi, A., Darwish, H., ... & Saeed, S. (2024). Water Deprivation and Sowing Times Alter Plant–Pollination Interactions and Seed Yield in Sunflower, Helianthus annuus L.(Asteraceae). Plants, 13(22), 3194.
L74-77: Revise the sentence: Over the years, European initiatives, such as the Council Directive 92/43/EEC (21 May 1992)... European strategies is not going well with the sentence.
Too much policy discussion. Try reducing it.
L87: Correct the in-text reference mistake.
L100-101: Add some recent studies here. For example:
1. Teixeira, T. S. M., Berggren, Å., & Riggi, L. G. A. (2025). Semi-natural habitat cover but not late season mass-flowering crops affect pollinator-plant networks in non-crop habitats. Agriculture, Ecosystems & Environment, 381, 109455.
General comment for introduction section: Revise the introduction to focus more on the scientific concepts of the study, including ecological principles and pollinator dynamics, rather than focusing more on policy frameworks. This modification will certainly provide a stronger context for the research question and its importance.
L128-131: Provide geographic coordinates.
L137: Correct intext citation.
L138, 140, 141: Give space between the value and mm.
L144-145: Revise the figure caption so it stands on its own. It should give enough context and details, so readers don’t have to hunt around the paper to figure out what’s going on.
L155-156: Same as above.
L146: How’s a new reader supposed to know what is Universal Transverse Mercator (UTM)? So, give a quick explanation with the abbreviation so they’re not left scratching their head.
L147: 1 km × 1 km
L157: Only for sampling wild bees?
L158: These aren’t cups; you can call them bowls to match their description more accurately. Be consistent!
L159: Why did you pick these specific colors for the pan traps? Was there a scientific reason or reference for choosing them? If yes, make sure to write it down! Readers are always interested to know the reason behind the color selection for the pan traps.
L162 and throughout manuscript: Make sure to add a space between the number and the unit. I won’t mention this again, so please check everywhere. Be consistent!
L165: Triplets of what?
L166: detergent details?
L197: non-significative?
L203: Just use: Bray-Curtis dissimilarity
L219: Which NMDS?
L190-224: Remember, your target audience is a new reader who is reading your paper to learn something new. Keep the language simple and straightforward so they don’t get lost in technical terms.
L227 and throughout manuscript where you’ve put brackets like this: No need to capitalize solitary bees and bumblebees.
Overall tables: Try to revise the tables as per MDPI format. The current tables are not as per the format.
Table 4: Add families.
L401-497: Skip the headings in the discussion and let it flow as a natural discussion. Also, add more recent studies to back up your points. I don’t see many up-to-date references. Add following studies:
1. Diffendorfer, J. E., Botello, F., Drummond, M. A., Ancona, Z. H., Corro, L. M., Thogmartin, W. E., ... & Sánchez-Cordero, V. (2024). Changes in landscape and climate in Mexico and Texas reveal small effects on migratory habitat of monarch butterflies (Danaus plexippus). Scientific Reports, 14(1), 6703.
2. Feigs, J. T., Huang, S., Holzhauer, S. I., Brunet, J., Diekmann, M., Hedwall, P. O., ... & Naaf, T. (2024). Bumblebees mediate landscape effects on a forest herb's population genetic structure in European agricultural landscapes. Ecology and Evolution, 14(7), e70078.
3. Filipiak, M. (2024). Plants other than animal-pollinated herbs provide wild bees with vital nutrients. Global Ecology and Conservation, 52, e02984.
4. Haider, S., Khan, F. Z., Gul, H. T., Ali, M., & Iqbal, S. (2024). Assessing the role of conservation strips in enhancing beneficial fauna in the wheat-cotton agricultural system in Punjab, Pakistan. Pak. J. Zool, 56, 1-9.
5. Karnchananiyom, S., Wayo, K., Sritongchuay, T., Warrit, N., Attasopa, K., & Bumrungsri, S. (2024). Local and landscape context affects bee communities in mixed fruit orchards in Southern Thailand. Agricultural and Forest Entomology, 26(1), 70-80.
6. Marrero, H. J., Gómez‐Martínez, C., Allasino, M. L., Haedo, J. P., González‐Estévez, M. A., & Lázaro, A. Local and landscape effects on the reproduction of wild bees and wasps in Mediterranean communities along a gradient of land‐use. Ecological Entomology.
7. Nolen, Z. J., Rundlöf, M., & Runemark, A. (2024). Species-specific erosion of genetic diversity in grassland butterflies depends on landscape land cover. Biological Conservation, 296, 110694.
8. Prober, S. M., Liedloff, A. C., England, J. R., Mokany, K., Ogilvy, S., & Richards, A. E. (2025). Accounting for the biodiversity benefits of woody plantings in agricultural landscapes: A global meta-analysis. Agriculture, Ecosystems & Environment, 381, 109453.
Author Response
Comment 1: L1-4: Consider revising the title to make it clear and concise.
Response 1: Thank you for pointing this out. We agree with this suggestion. Therefore, we have revised the title to make it more concise. Revised Title: “Biodiversity-Friendly Management in Olive Groves Supports Pollinator Conservation in a Mediterranean Terraced Landscape.”
Comment 2: L16-26: It is recommended to add pan traps and observation walks as methods to improve the summary.
Response 2: Agree, the sentence in the Simple Summary has been modified. Revised sentence L18-21: “The research assessed wild bees, flowers, and butterfly populations across three habitat types: olive groves, herbaceous linear elements, and woody areas, using pan-traps and observation walks over two years.”
Comment 3: L38-40: Do you want to say analysis using GLMMs?
Response 3: Agree, the sentence is clearer that way. Revised L37: “Analysis showed that”
Comment 4: L27-48: Check the maximum number of words allowed as per the MDPI format and make it according to the guidelines.
Response 4: Thank you for pointing this out, the maximum length is “about 200 words” (https://www.mdpi.com/journal/insects/instructions#preparation), and this abstract used to be 245 words long. The overall structure of the abstract has been revised, and it is now counting 221 words.
Comment 5: L49: Expand the keywords and consider including the region and farming system.
Response 5: Thanks for suggesting the expansion of the keywords. “Mediterranean landscape” and “olive growing” have been included.
Comment 6: L54: Add family.
Response 6: Thank you for pointing this out. We agree that adding the family provides more clarity and scientific precision. Therefore, we have updated the text to include the relevant family. The updated text is L52 and L53. “(Imenoptera: Apidae)” and “(Lepidoptera: Rhopalocera)”
Comment 7: L58: Add some recent papers here. For example:
1. Straub, F., Birkenbach, M., Boesing, A. L., Manning, P., Olsson, O., Kuppler, J., ... & Ayasse, M. (2025). Local and landscape factors differently influence health and pollination services in two important pollinator groups. Science of The Total Environment, 959, 178330. 2. Ali, Q., Ali, M., Khan, F. Z. A., Noureldeen, A., Alghamdi, A., Darwish, H., ... & Saeed, S. (2024). Water Deprivation and Sowing Times Alter Plant–Pollination Interactions and Seed Yield in Sunflower, Helianthus annuus L.(Asteraceae). Plants, 13(22), 3194.
Response 7: Thank you for the insightful suggestion, the Straub et al. paper has been included at L66 because of its inherence with the intensity of land-use.
Comment 8: L74-77: Revise the sentence: Over the years, European initiatives, such as the Council Directive 92/43/EEC (21 May 1992)... European strategies is not going well with the sentence.
Response 8: The sentence has been thoroughly revised based on the suggestion at L76, together with the whole introduction.
Comment 9: Too much policy discussion. Try reducing it.
Response 9: The part about policies has been reduced. More than 10 lines have been removed.
Comment 10: L87: Correct the in-text reference mistake.
Response 10: The reference has been removed in the attempt of shortening the section about policies
Comment 11: L100-101: Add some recent studies here. For example: Teixeira, T. S. M., Berggren, Å., & Riggi, L. G. A. (2025). Semi-natural habitat cover but not late season mass-flowering crops affect pollinator-plant networks in non-crop habitats. Agriculture, Ecosystems & Environment, 381, 109455.
Response 11: Thank you for the suggestion, the paper has been included in the citation at L63.
Comment 12: General comment for introduction section: Revise the introduction to focus more on the scientific concepts of the study, including ecological principles and pollinator dynamics, rather than focusing more on policy frameworks. This modification will certainly provide a stronger context for the research question and its importance.
Response 12: Thank you for this precious suggestion, the whole introduction has undergone major revisions also taking into consideration the suggestions from other reviewers.
Comment 13: L128-131: Provide geographic coordinates.
Response 13: Coordinates have been provided at L128.
Comment 14: L137: Correct intext citation.
Response 14: Correction applied.
Comment 15: L138, 140, 141: Give space between the value and mm.
Response 15: Space given, here and elsewhere.
Comment 16: L144-145: Revise the figure caption so it stands on its own. It should give enough context and details, so readers don’t have to hunt around the paper to figure out what’s going on.
Response 16: Thank you for the suggestion; all captions have been rephrased for increased clarity.
Comment 17: L155-156: Same as above.
Response 17: Thank you for the suggestion; all captions have been rephrased for increased clarity.
Comment 18: L146: How’s a new reader supposed to know what is Universal Transverse Mercator (UTM)? So, give a quick explanation with the abbreviation so they’re not left scratching their head.
Response 18: A brief explanation has been provided at L147-150
On behalf of all the authors, thank you for your insightful suggestions and for the proposed references, which proved very useful.
Reviewer 2 Report
Comments and Suggestions for Authors
The manuscript “The contribution of biodiversity-friendly vegetation management in olive Groves Supports the conservation of pollinating insects in a typical Mediterranean terraced landscape” is fascinating in olive crops and their surrounding areas. Integrating the type of land use around crops is very important and in the studies before neglected. Therefore, this manuscript has a strength in considering the surrounding land use areas. The manuscript is well written in general the statistical methods are adequate. However, some points have not been correctly assessed as the studies in heterogenous areas for bees, some presentation and statistical points that should be assessed to publish the Ms.
The hypothesis is that areas with a greater diversity of plants will boost insect diversity, but there is not an adequate introduction talking specifically about these findings in previous studies. So it is not suddenly a bit abrupt that this hypothesis is presented. Please add something about this.
Several studies suggest that flower-rich sites with intermediate levels of perturbation but heterogenous by human activities (e.g., seminatural areas, human settlements, and agriculture) can have a positive effect on bee and fly abundance and the total diversity of insect pollinators (Kratschmer et al., 2019; Aguilera et al., 2020; Meng et al., 2012; Landaverde-González et al., 2017; Cusser et al., 2019; Escobedo-Kenefic et al., 2020; Coutinho et al., 2021).
Aguilera, G., Roslin, B., Miller, K., Tamburini, G., Birkhofer, K., Caballero-Lopez, B., et al. (2020). Crop diversity benefits carabid and pollinator communities in landscapes with semi-natural habitats. J. Appl. Ecol. 57, 2170–2179. doi: 10.1111/1365-2664.13712
Cusser, S., Grando, C., Zucchi, M. I., López-Uribe, M. M., Pope, N. S., Ballare, K., et al. (2019). Small but critical: semi-natural habitat fragments promote bee abundance in cotton agroecosystems across both Brazil and the United States. Landsc. Ecol. 34, 1825–1836. doi: 10.1007/s10980-019-00868-x
Coutinho, J. G. E., Hipólito, J., Santos, R. L. S., Moreira, E. F., Boscolo, D., and Viana, B. F. (2021). Landscape structure is a major driver of bee functional diversity in crops. Front. Ecol. Evol. 9, 624835. doi: 10.3389/fevo.2021.624835
Kratschmer, S., Pachinger, B., Schwantzer, M., Paredes, D., Guzmán, G., Gómez, J. A., et al. (2019). Response of wild bee diversity, abundance, and functional traits to vineyard inter-row management intensity and landscape diversity across Europe. Ecol. Evol. 9, 4103–4115. doi: 10.1002/ece3.5039
Landaverde-González, P., Quesada-Euán, J., Theodorou, P., Murray, T., Husemann, M., Ayala, R., et al. (2017). Sweat bees on hot chilies: provision of pollination services by native bees in traditional slash-and-burn agricultural in Yucatán Peninsula of Tropical Mexico. J. Appl. Ecol. 54, 1814–1824. doi: 10.1111/1365-2664.12860
Meng, L. Z., Martin, K., Liu, J. X., Burger, F., and Chen, J. (2012). Contrasting responses of hoverflies and wild bees to habitat structure and land use change in a tropical landscape (southern Yunnan, SW China). Insect Sci. 19, 666–676. doi: 10.1111/j.1744-7917.2011.01481.x
Escobedo-Kenefic, N., Landaverde-González, P., Theodorou, P., Cardona, E., Dardón, M. J., Martínez, O., et al. (2020). Disentangling the effects of local resources, landscape heterogeneity and climatic seasonality on bee diversity and plant-pollinator networks in tropical highlands. Oecologia 194, 333–344. doi: 10.1007/s00442-020-04715-8
The structure of the manuscript is generally well-organized; however, the results and discussion sections become somewhat disorganized. For instance, you discuss aspects such as the color of plants and pan traps, which are not directly related to your hypotheses. This inclusion adds confusion about your primary objectives. Additionally, the first part of the introduction focuses more on caveats than on presenting and summarizing your goals, which further muddles the manuscript’s clarity. I recommend focusing strictly on your goals and maintaining a structure that consistently supports them.
Regarding statistics: You mention that random factors accounted only for spatial variation, but both year and time were significant. How can you be confident about differences between T1-T4 if you are not accounting for the variation introduced by year? You can only assert these differences between times if you control for a year or clarify how this analysis was conducted. Did you have eight-time points? Similarly, other analyzed variables might also be affected by inter-annual differences. For this reason, the year should be included as a variable to control for its effects.
The specific details are the next:
L70. Intensive agriculture has been observed to affect diversity at the local scale (alpha diversity), while at beta and gamma diversity the effect is not always so pronounced (Tasng et al., 2025). Please add this to the introduction to give a good argument why is necessary to study olive crop systems.
Tsang, TPN., De Santis, AA., Armas-Quiñonez, AG. et al. 2024. Land use change consistently reduces α but not β and γ diversity of bees. Glob Change Biol, 31: (1) e70006. https://doi.org/10.1111/gcb.70006
L119-120. You need to introduce more deeply the diversity of floral resources coming from seminatural heterogeneous areas, otherwise, it feels a little like coming from nowhere. You mention something but exclusively for olive crops and briefly, go deeper. Similarly, you have to consider this for the discussion.
L121-125. this comparison between bees and butterflies was not observed. For this hypothesis, I would expect to see some comparison of the sites and times in which bees and butterflies have peaks and base the discussion on this. Do they differ? I don’t know. I don’t see any comparative figure.
L180-182. so, each time represents a month? there is the problem of the year though. Or did you do different times for 2022 and 2021 there should be then T7 and not just T4. If you want to maintain just four times you need to control for "year" as we already observed that there was a significant difference from one year to the other.
L195. Why "Year"was not analysed as random? especially when you analyse times. As year was significant you should put that apart as you put the group together
L203. why Bray Curtis index? did you measure the fit of the different metrics to your data? I mean can be right. But we need to see some arguments regarding the fit of the index.
L226. you begin in 3.2 with butterflies, but here you just said Pant traps when you should focus on bees. In addition, you are focused on the insects, not on the method. This makes your nice paper look messy from here on.
L227-228. detail of the bees sampled is necessary
L254. supplementary info is not clear, there are no tables but Excel files. In addition, on the pan traps Excel page, there is a mix of insects and plants. So, I am not sure what are you showing. It will be nicer that at least you show a table just with the insect sampled and a graph showing abundances.
L280, L307, L364. L384. and each of the figures and tables has to be described in a form that you don’t need to go back to the text. Having said that, you just have one figure where you indicate the code for T1 to T4 and for OL, HL, and WA. In this form, the readers have to remember or check the Ms. This is uncomfortable. I recommend that you indicate in every table or figure or that you refer to the code that is clearly stated in one figure or table.
L288-289. Are the Times referring to the mid of the month? You said mid-May, mid-june, but I understood you were referring to the whole month. Stick to one terminology and don’t change it otherwise, you cause confusion.
L321-332. I was understanding that this was not a hypothesis. How is colour related with composition of bees and butterflies and the effect of the habitat on them?
L350. where is the table and image with this information? The information on the excel files is not the same as the amounts you present here for bees and butterflies.
L403-416. this is not part of your goals or hypotheses and I understand that these are caveats that you need to discuss, but should be later and not as the first part. In the first part of the discussion, you need to summarize your hypothesis and how the data are related to them. It is not obligatory but is a practice that helps you to make the Ms straightforward
L417. bee like? I was understanding that you have wild bees and Bombus. Although it is very difficult to know as you don't show in detail the tables. Do you also account for bee-like flies? Is what you mean?
L426. of course you need to discuss which kind of floral resources. Abundant resources for the same dominant species are not the same as abundant heterogeneous floral resources, see Landavere et al., 2017. Landaverde-González, P., Quesada-Euán, J., Theodorou, P., Murray, T., Husemann, M., Ayala, R., et al. (2017). Sweat bees on hot chilies: provision of pollination services by native bees in traditional slash-and-burn agricultural in Yucatán Peninsula of Tropical Mexico. J. Appl. Ecol. 54, 1814–1824. doi: 10.1111/1365-2664.12860
Besides in methods you need to describe a little more the composition of each area so you can discuss it better here.
L425-426. this difference is just one month. Besides, could be that there are still a lot of flowers
L450. Do you mean June?
L452. how was the precipitation in these months? It has been observed that rain can boost blooming and therefore pollinators.
Casia-achje et al., 2024
Casiá Ajché, Q.B., Escobedo-Kenefic, N., Escobar, D.D, Cardona, E., Mejía, A., Morales, X., Enríquez, E. Landaverde-Gonzalez, P. 2024. Unveiling the effects of land use and intra-seasonality on bee and plant diversity and their ecological interactions in vegetation surrounding coffee plantations Frontiers in Bee Science, 2, 1408854. https://doi.org/10.3389/frbee.2024.1408854
L454. which specific environmental conditions. You have to be clear to understand better the patterns you are observing
L464. reference for this last sentence.
L476-478. this is what he said, how is related to your data?
L491-493. this is also true for bees but you didn't discuss it
Author Response
Comment 1: The hypothesis is that areas with a greater diversity of plants will boost insect diversity, but there is not an adequate introduction talking specifically about these findings in previous studies. So it is not suddenly a bit abrupt that this hypothesis is presented. Please add something about this.
Response 1: Thank you for pointing this out. The part of the Introduction leading up to the hypotheses has been majorly revised and new references (thank you again for suggesting them) have been added. Specifically, L114-122 now introduces the proposed hypotheses with more context from previous research.
Comment 2: The structure of the manuscript is generally well-organized; however, the results and discussion sections become somewhat disorganized. For instance, you discuss aspects such as the color of plants and pan traps, which are not directly related to your hypotheses. This inclusion adds confusion about your primary objectives. Additionally, the first part of the introduction focuses more on caveats than on presenting and summarizing your goals, which further muddles the manuscript’s clarity. I recommend focusing strictly on your goals and maintaining a structure that consistently supports them.
Response 2: The short summary, abstract, and introduction have undergone major revisions. I hope they look better now. Color of flowers may be a possible explanation, besides habitat, for pollinators presence or lack thereof (Reverté S, Retana J, Gómez JM, Bosch J. Pollinators show flower color preferences but flowers with similar colours do not attract similar pollinators. Ann Bot. 2016 Aug;118(2):249-57. doi: 10.1093/aob/mcw103. Epub 2016 Jun 20. PMID: 27325897; PMCID: PMC4970366.). The proposed colour analysis is very basic in its nature and I agree it is not part of the hypotheses of the study. This part has been removed.
Comment 3: Regarding statistics: You mention that random factors accounted only for spatial variation, but both year and time were significant. How can you be confident about differences between T1-T4 if you are not accounting for the variation introduced by year? You can only assert these differences between times if you control for a year or clarify how this analysis was conducted. Did you have eight-time points? Similarly, other analyzed variables might also be affected by inter-annual differences. For this reason, the year should be included as a variable to control for its effects.
Response 3: Thank you for signaling this, a detailed response is provided at Response 7.
Comment 4: L70. Intensive agriculture has been observed to affect diversity at the local scale (alpha diversity), while at beta and gamma diversity the effect is not always so pronounced (Tsang et al., 2025). Please add this to the introduction to give a good argument why is necessary to study olive crop systems. Tsang, TPN., De Santis, AA., Armas-Quiñonez, AG. et al. 2024. Land use change consistently reduces α but not β and γ diversity of bees. Glob Change Biol, 31: (1) e70006. https://doi.org/10.1111/gcb.70006
Response 4: Thank you for the suggestion of this nice paper, it has been included in the introduction as a reference at L68-72: " Intensive agricultural practices have been recently shown to reduce bees’ α-diversity (i.e. local diversity within a site) consistently across taxa, while their effects on β-diversity (i.e. compositional differences between sites) and γ-diversity (i.e. overall diversity across the landscape) remain less clearly understood [9].”
Comment 5: L119-120. You need to introduce more deeply the diversity of floral resources coming from seminatural heterogeneous areas, otherwise, it feels a little like coming from nowhere. You mention something but exclusively for olive crops and briefly, go deeper. Similarly, you have to consider this for the discussion.
Response 5: Several of the suggested references (Kratschmer et al., 2019; Aguilera et al., 2020; Meng et al., 2012; Landaverde-González et al., 2017; Cusser et al., 2019; Escobedo-Kenefic et al., 2020; Coutinho et al., 2021) (thank you again) have been introduced at L115. The discussion has been thoroughly revised in the attempt to provide more robust insights.
Comment 6: L121-125. this comparison between bees and butterflies was not observed. For this hypothesis, I would expect to see some comparison of the sites and times in which bees and butterflies have peaks and base the discussion on this. Do they differ? I don’t know. I don’t see any comparative figure.
Response 6: Thank you for pointing this unclearness out. The word “differ” has been changed with the word “vary” at L120, highlighting the fact that the manuscript does not compare bees and butterflies directly, but rather the patterns in their responses to habitats and sampling time
Comment 7: L180-182. so, each time represents a month? there is the problem of the year though. Or did you do different times for 2022 and 2021 there should be then T7 and not just T4. If you want to maintain just four times you need to control for "year" as we already observed that there was a significant difference from one year to the other.
Response 7: Thank you for your observation. Each time (T1-T4) represents a specific month, with the same sampling protocol repeated across two years (2021 and 2022), although only T1 and T3 were assessed in 2021 for pan traps, and only T2, 3, 4 were assessed in 2021 for observation walks. To address inter-annual differences, we included "Year" as a fixed effect in the statistical model, ensuring that any significant variation between years is explicitly accounted for. The GLMM used for the analysis is specified as follows: glmmTMB(SolBumb ~ Habitat + Year + Time + (1 | Landscape), family = nbinom2()). We initially tested the Year:Time interaction but found it to be non-significant, and furthermore as not all times were sampled in both years, its inclusion would probably hinder the validity of the model as it would make it rank-deficient.
We assume that the effect of time (months) on the response variable was consistent across years. The inclusion of Year as a random effect did minimally increase AIC and did not change the post-hoc tests’ results, so the simpler model has been preferred. Time was not treated as nested within Year because it represents a shared temporal structure (T1-T4, corresponding to the same four months) across both years (although for 2021 only T1, T2 and T3 were sampled for observation plots and T1 and T3 for pantraps). Nesting would imply that the months are unique to each year (which may make sense because of climatic conditions), which for simplicity we consider not to be the case in this study. Instead, we controlled for inter-annual differences directly by including "Year" as a fixed effect. This approach preserves the interpretability of temporal patterns (e.g., seasonal trends) across years while accounting for any baseline differences between 2021 and 2022, which indeed proved significant (p = 0.04 with Hochberg’s adjustment. An explanation is provided at L211-214
Comment 8: L195. Why "Year"was not analysed as random? especially when you analyse times. As year was significant you should put that apart as you put the group together
Response 8: See Response 7. The results would have looked identical in their interpretation. A brief explanation of the choice has been added to the caption of Figure 3 at L274 Solitary bee and bumblebee abundances were aggregated across the two sampling years (2021 and 2022). Inter-annual differences were controlled for by including Year as a fixed effect in the model”
Comment 9: L203. why Bray Curtis index? did you measure the fit of the different metrics to your data? I mean can be right. But we need to see some arguments regarding the fit of the index.
Response 9: The Bray Curtis index is the most used index in ecological studies and specifically suitable for abundance data (our data are abundance data). To ensure the appropriateness of the index, we also compared the fit of different metrics (e.g., Jaccard (only presence/absence values, Euclidean) to our data using stress values from NMDS. The Bray-Curtis index consistently yielded lower stress values, indicating a better fit for ordination. Stress values from our NMDS are reported in the captions (L339 and L399) of the figures with NMDS plots and at L335 and 395. They are a bit high, but within the acceptable range. Details on the NMDS have been added at L226-229.
Comment 10: L226. you begin in 3.2 with butterflies, but here you just said Pant traps when you should focus on bees. In addition, you are focused on the insects, not on the method. This makes your nice paper look messy from here on.
Response 10: Thank you for pointing this out, the heading has been modified and it now states “Wild bees” at L241.
Comment 11: L227-228. detail of the bees sampled is necessary
Response 12: Taxonomic definition of what we consider wild bees has been added at L231
Comment 12: L254. supplementary info is not clear, there are no tables but Excel files. In addition, on the pan traps Excel page, there is a mix of insects and plants. So, I am not sure what are you showing. It will be nicer that at least you show a table just with the insect sampled and a graph showing abundances.
Response 12: Agree. Basic plots and tables have been provided in the new Supplemetary material.
Comment 13: L280, L307, L364. L384. and each of the figures and tables has to be described in a form that you don’t need to go back to the text. Having said that, you just have one figure where you indicate the code for T1 to T4 and for OL, HL, and WA. In this form, the readers have to remember or check the Ms. This is uncomfortable. I recommend that you indicate in every table or figure or that you refer to the code that is clearly stated in one figure or table.
Response 13: Agree, explanations for abbreviations have been added to each table and figure that presents them together with more thorough explanations of the presented data.
Comment 14: L288-289. Are the Times referring to the mid of the month? You said mid-May, mid-june, but I understood you were referring to the whole month. Stick to one terminology and don’t change it otherwise, you cause confusion.
Response 14: Thank you for pointing at this ambiguity. Each time represent a month. The terminology has been set accordingly in every section and caption of the manuscript except the materials and methods where the exact dates of the samplings have been reported.
Comment 15: L321-332. I was understanding that this was not a hypothesis. How is colour related with composition of bees and butterflies and the effect of the habitat on them?
Response 15: Agree, the color analysis has been removed from the manuscript.
Comment 16: L350. where is the table and image with this information? The information on the excel files is not the same as the amounts you present here for bees and butterflies.
Response 16: Thank you for signaling this. I have double checked the total butterfliy abundance and it sums up to 1023 specimens. Table S1 in the supplementary materials also confirms this total for butterflies.
Comment 17: L403-416. this is not part of your goals or hypotheses and I understand that these are caveats that you need to discuss, but should be later and not as the first part. In the first part of the discussion, you need to summarize your hypothesis and how the data are related to them. It is not obligatory but is a practice that helps you to make the Ms straightforward.
Response 17: Thank you for reporting this, I would generally agree, but I prefer to highlight these caveats at the stsrt of the discussion to allow the reader to take this into account when I discuss the results.
Comment 18: L417. bee like? I was understanding that you have wild bees and Bombus. Although it is very difficult to know as you don't show in detail the tables. Do you also account for bee-like flies? Is what you mean?
Response 18: Thank you for this correction, you are right these are solitary bees and bumblebees, not hoverflies. The exact taxonomic definition of what is considered what has been included in the materials and methods and in the results, and the group has been referred to as “solitary bees and bumblebees” or “hymenopteran pollinators” across the Ms.
Comment 19: L426. of course you need to discuss which kind of floral resources. Abundant resources for the same dominant species are not the same as abundant heterogeneous floral resources, see Landavere et al., 2017. Landaverde-González, P., Quesada-Euán, J., Theodorou, P., Murray, T., Husemann, M., Ayala, R., et al. (2017). Sweat bees on hot chilies: provision of pollination services by native bees in traditional slash-and-burn agricultural in Yucatán Peninsula of Tropical Mexico. J. Appl. Ecol. 54, 1814–1824. doi: 10.1111/1365-2664.12860
Response 19: Thank you for suggesting these references to further contextualize the findings of the Ms. The suggested reference has been added to the discussion at L442. It is an interesting suggestion to evaluate floral resources in general, but in this study we did not characterize flower communities functionally (i.e. with traits), so any discussion in this regard would be poorly supported by data. The color analysis could have proven useful here, but its quality was questionable (taking the color of the most abundant species is probably too simplistic), so it was dropped.
Comment 20: L425-426. this difference is just one month. Besides, could be that there are still a lot of flowers
Response 20: I acknowledge that the temporal difference between sampling times is only one month, and it is possible that floral resources remain abundant during this period. However, as the interaction between time of sampling and habitat was not significant, the observed trends suggest that the effects of habitat on pollinator abundance are consistent across the sampling period, irrespective of potential short-term fluctuations in flower availability. L448-450 tries to clarify this.
Comment 21: L450. Do you mean June?
Response 21: Thank you for noticing this, indeed yes I was referring to June. The sentence was redundant and it has been removed.
Comment 22: L452. how was the precipitation in these months? It has been observed that rain can boost blooming and therefore pollinators. Casia-achje et al., 2024 Casiá Ajché, Q.B., Escobedo-Kenefic, N., Escobar, D.D, Cardona, E., Mejía, A., Morales, X., Enríquez, E. Landaverde-Gonzalez, P. 2024. Unveiling the effects of land use and intra-seasonality on bee and plant diversity and their ecological interactions in vegetation surrounding coffee plantations Frontiers in Bee Science, 2, 1408854. https://doi.org/10.3389/frbee.2024.1408854
Response 22: Thank you for the question and for the useful reference. As shown in Figure 1, the precipitation was low during summer 2021 and 2022. A short sentence about this has been added at L141. The suggested reference has been added in the discussion about solitary bees and bumblebees at L454.
Comment 23: L454. which specific environmental conditions. You have to be clear to understand better the patterns you are observing
Response 23: Thank you for highlighting this unclear sentence. It has been changed at L476: “in response to specific conditions, both resource (food, nesting) and biotope-related”
Comment 24: L464. reference for this last sentence.
Response 24: A reference has been added at L484 (Rundlöf M, Smith HG (2006) The effect of organic farming on butterfly diversity depends on landscape context. Journal of Applied Ecology 43:1121–1127. https://doi.org/10.1111/j.1365-2664.2006.01233.x)
Comment 25: L476-478. this is what he said, how is related to your data?
Response 25: The sentence has been thoroughly rephrased at L492-onwards, and more references have been added.
Comment 26: L491-493. this is also true for bees but you didn't discuss it
Response 26: Thank you for pointing out that, the correct word here is “pollinators”, not “butterflies”. The correction has been applied at L499.
Dear Reviewer 2, I sincerely wish to thank you for your in depth reading and insightful commenting on the manuscript. Most if not all of your comments proved very useful, constructive, and well defined. I think this manuscript has generally improved mostly thank to your suggestions. I appreciate and value the time and effort you put in this regard. Thank you.
Reviewer 3 Report
Comments and Suggestions for Authors
Dear authors,
I was pleased to review your ms entitled "he contribution of biodiversity-friendly vegetation management in olive groves supports the conservation of pollinating insects in a typical Mediterranean terraced landscape" which investigated the influence of habitat on flower and pollinator communities.
While the ms is interesting, the study has some major flaws. The first is that the alpha diversity such as species richness is not evaluated especially for Butterfly as a explained variable (= Y) in your GLMMs. What about species richness ? I think you could improve this and refont your summary, M&M, Results and discussion according to these changes. For the wild bees, only the abundance is taken in account. Why not identify them to species level to have the species richness of wild bee as information. I think that could be capital for your paper. You also tell that the habitat diversity and floral resources have influence on the pollinator communities but you did not test it, you just test the floral and pollinator communities by habitats and sampling times solely on univariate way. You can also did it on a multivariate way by using co-inertia analysis for example to test the influence of the floral Community on the Butterfly Community. In the result part, you presented your datasets too briefly. We do not know how your taxonomic datasets are distributed in term of family or genera. What are main species of your study ? For all the point 3.1. you can summarize your result by being more factual, be straight to the point with your result, there are too much and you lost the readers. Please focus on the main ones. You results looks more like a report than a scientific paper.
Specific remarks:
38: You are not mandatory to call your GLMM in the abstract. GLMM are a just a mean to explain your story, not your story.
54: What do you mean by primarly ? Is it possible that olives can be insect pollinated ? In this case by which pollinator guild ?
60: What do you mean by ecological balance ?
64: If you refer to several species please use "Community" instead of "populations", to check in all the text.
Figure 1 and L135 to 142: In which case is it relevant for your study to know the amount of precipitation ? Do you test it ? For me, just stupilating the climate in the region which is hot-summer Mediterranean is enough
151-152: Only abundance and not diversity or species richness ?
187: On which morphological keys do you support your identifications please add the reference or the reference collection.
Author Response
Comment 1: While the ms is interesting, the study has some major flaws. The first is that the alpha diversity such as species richness is not evaluated especially for Butterfly as a explained variable (= Y) in your GLMMs. What about species richness? I think you could improve this and refont your summary, M&M, Results and discussion according to these changes. For the wild bees, only the abundance is taken in account. Why not identify them to species level to have the species richness of wild bee as information. I think that could be capital for your paper. You also tell that the habitat diversity and floral resources have influence on the pollinator communities but you did not test it, you just test the floral and pollinator communities by habitats and sampling times solely on univariate way. You can also did it on a multivariate way by using co-inertia analysis for example to test the influence of the floral Community on the Butterfly Community. In the result part, you presented your datasets too briefly. We do not know how your taxonomic datasets are distributed in term of family or genera. What are main species of your study ? For all the point 3.1. you can summarize your result by being more factual, be straight to the point with your result, there are too much and you lost the readers. Please focus on the main ones. You results looks more like a report than a scientific paper.
Response 1: Thank you for the valid suggestions. The species richness of butterflies has been found highly colinear with butterfly abundance (R=0.86) and for this reason, we decided to only focus on abundance. Furthermore, as the environmental sampling is very basic (only habitat and time of sampling), we think that delving into further complexities would lead to more vague conclusions than those we managed to obtain with this work. The bee identification was planned, but due to logistical constraints we were forced to skip it. Floral community was sampled together with pan traps, and in the radius of 2 m from each bowl, so the flower data is only applicable to hymenopteran pollinators as the transects for butterfly observations, although consistent with the pan traps’ transects, focused on a broader area. The multivariate analysis we performed, PERMANOVA and NMDS, have the less strict assumptions and in our opinion work well for the scope of the study: highlighting the relative importance of all habitats in the traditional Mediterranean olive growing landscapes.
The dataset has been more in depth presented in the new Supplementary materials, with tables and boxplot.
Overall, the manuscript has undergone major changes in order to improve its readability. The analysis on flower colors has been dropped in the attempt to focus more on the hypotheses, which together with the introduction and the discussion have been added several new references.
Comment 2: L38: You are not mandatory to call your GLMM in the abstract. GLMM are a just a mean to explain your story, not your story.
Response 2: Agree. L38 now has “analysis” instead of “GLMM”
Comment 3: L54: What do you mean by primarly ? Is it possible that olives can be insect pollinated ? In this case by which pollinator guild ?
Response 3: Thank you for the question, there are some cultivars that self-pollinate, while others benefit from insect pollination (G.C. Koubouris, C.M. Breton, I.T. Metzidakis, M.D. Vasilakakis, Self-incompatibility and pollination relationships for four Greek olive cultivars, Scientia Horticulturae, Volume 176, 2014, Pages 91-96, ISSN 0304-4238, https://doi.org/10.1016/j.scienta.2014.06.043.) The reference has been added at L52.
Comment 4: 60: What do you mean by ecological balance ?
Response 4: Thank you for the question, ecological balance is intended here as a dynamic stable state between all the organisms in the ecosystem, but for added clarity the sentence has been rephrased at L58: “pollinators represent a key element of the agroecosystem”.
Comment 5: L64: If you refer to several species please use "Community" instead of "populations", to check in all the text.
Response 5: Thank you for the suggestion, all occurrences of “populations” have been replaced with “community” throughout the manuscript.
Comment 6: Figure 1 and L135 to 142: In which case is it relevant for your study to know the amount of precipitation ? Do you test it ? For me, just stupilating the climate in the region which is hot-summer Mediterranean is enough
Response 6: Thank you for the question, some more insightful information about the climate is now provided at L141-143 and L452.
Comment 7: L151-152: Only abundance and not diversity or species richness ?
Response 7: Yes, pan traps were used only for abundance of wild bees and bumblebees because we were not able to identify the specimens in the traps.
Comment 8: L187: On which morphological keys do you support your identifications please add the reference or the reference collection.
Response 8: Thank you for noticing the lack of this information, it has been provided at L196.
On behalf of the authors, thank you for your suggestions and insights.
Round 2
Reviewer 1 Report
Comments and Suggestions for Authors
The authors have made an excellent effort to improve the draft. Well done.
Author Response
Dear Reviewer 1, thank you for your appreciation and thank you again for the useful suggestions you provided at Round 1.
Best regards
The authors
Reviewer 2 Report
Comments and Suggestions for Authors
The manuscript "Biodiversity-Friendly Management in Olive Groves Supports Pollinator Conservation in a Mediterranean Terraced Landscape" has been improved. I want to congratulate the authors for their effort.
There are still some points that are not clear.
L207-208. As the year is significantly different you have to control for its variation. If you put it as a fixed variable, it is just an explanatory (independent) variable, but the effect of sampling in different years is still not accounted for. For example in your answer: "interannual differences were controlled for by including Year as a fixed effect in the model” In this form you are not controlling for year, you are just checking its explanatory power. Do you understand the mistake here? you are probably finding differentiation that is caused due to the difference in year and not due to habitat and time alone. To determine this you need to put Year as random in the analysis.
olBumb ~ Habitat + Time + (1 | Landscape + 1 |Year )
By the other side it is not Landscape in some form related to habitat?
In your response, you mention that you repeated the analysis and you did not find difference, but this is not explained in this part of the methods. Besides in many answers to your response letter you continue with this argument, that sometimes is confuse.
For example with this sentence, you make it more urgent to control for year. "Nesting would imply that the months are unique to each year (which may make sense because of climatic conditions)"
L212-213. the interaction was not significant. But you still have year alone as significant affecting. I understand what you mean with check the explanatory value of year, including it as fixed, but is not your goal and it may affect the spatial variation. You should control for year, even if there is just one month repeated in the two years.
L327-332. And here is when I get a little more confused. I understand you check for year as random, but your argument in the response and in the discussion is to be included as fixed. However, in this table, you don't have year as fixed. So I am confused of what you were talking about.
L442.I think my previous comment was unclear. What I meant was that you need to recall your expectations so that you can quickly discuss whether they were met or not. I mean you should begin as something like. Regarding our first prediction about the hymenopteran pollinator abundance... and something brief and regarding our second prediction about composition of butterfly abundance would vary...
Then as you mention you can mention the caveats...
You are almost there, just make clear these points. Looking forward to the last revised version.
Reviewer 3 Report
Comments and Suggestions for Authors
The authors did a great job to improve their ms. But I always find that there is too many results and too many descriptions of the results which overflood the readers. I would prefer a summary of the most important and impactfull results in this part.
Author Response
Comment 1: The authors did a great job to improve their ms. But I always find that there is too many results and too many descriptions of the results which overflood the readers. I would prefer a summary of the most important and impactfull results in this part.
Response 1: Thank you for pointing out this problem, we tried our best to present our results in the most complete yet synthetical manner. We believe all paragraphs in the Results follow the same logical framework: providing all the elements for the reader to understand what was found in the analysis, in order to provide sound proofs for what is stated in the Discussion and in the Conclusions.
Furthermore, we have now thoroughly rethought the Discussion, which now encompasses the limitations of the study together with the reasonings on the results, and not at the beginning of the section. We hope this will improve the readability of the work.
Round 3
Reviewer 2 Report
Comments and Suggestions for Authors
There are no extra comments. I liked how the author responded to the points. Congratulations on the good contribution.